# Unraveling Neurodiversity: Insights from Neuroscientific Perspectives

**Hagar Goldberg**

Department of Psychology, University of British Columbia, Vancouver, BC V6T 1Z4, Canada; hagar.goldberg@ubc.ca

**Definition:** Neurodiversity is a concept and a social movement that addresses and normalizes human neurocognitive heterogeneity to promote acceptance and inclusion of neuro-minorities (e.g., learning disabilities, attention disorders, psychiatric disorders, and more) in contemporary society. Neurodiversity is attributed to nature and nurture factors, and about a fifth of the human population is considered neurodivergent. What does neurodiversity mean neuroscientifically? This question forms the foundation of the present entry, which focuses on existing scientific evidence on neurodiversity including neurodiversity between and within individuals, and the evolutional perspective of neurodiversity. Furthermore, the neuroscientific view will be synergistically integrated with social approaches, particularly in the context of the normalization of neurodiversity and its association with the medical and social models of disability. This multidimensional analysis offers a cohesive and comprehensive understanding of neurodiversity, drawing insights from various vantage points, such as social, psychological, clinical, and neuroscientific viewpoints. This integrated approach fosters a nuanced and holistic discussion on the topic of human diversity.

**Keywords:** neurodiversity; neurotypical; neurodivergent; neurominorities; neurodevelopmental

## 1. Different Perspectives on Neurodiversity

### 1.1. Neurodiversity as a Social Movement—Reclaiming the Right to Be Different

While neurodiversity may sound like a term from a brain science textbook, effectively, it emerged from a social justice movement and social science research [1]. The term neurodiversity was coined by sociologist Judy Singer in the late 1990s [2] to offer a new perspective on human variation in perception and communication styles. Singer drew inspiration from social movements fighting for the rights and needs of social minorities and environmental science emphasizing the benefits of biodiversity for a thriving ecosystem. Singer suggested that neurological diversity is a healthy and natural characteristic of the human species and that neurological differences are part of natural human diversity and should not automatically be pathologized.

Since the 1990s, the neurodiversity movement has been directed at raising awareness and appreciation for the diversity in human cognition and breaking down structures of exclusion. However, within the neurodiversity movement, there is an ongoing debate about what self-advocacy means and the aspired social and political outcomes. Some argue that neurological differences should be normalized and accepted. This voice is expressed, for example, in the words of John: "As an adult with autism, I find the idea of natural variation to be more appealing than the alternative—the suggestion that I am innately bad or broken and in need of repair... Asserting that I am different—not defective—is a much healthier position to take. Realizing the idea is supported by science is even better" [3].

Others argue against normalization and seek acknowledgment of the disability or illness related to their neuronal atypicality. An example of this approach is expressed in the words of Sue: "As a person who lives with autism daily and will not live a normal life, I find people who are high functioning and saying society should not look for a cure offensive" [4].

These two opposite approaches correspond (respectively) with the social and (classic) medical models of disability. According to the medical model, disability is caused by a dysfunction of the individual and requires treatment or a cure. The social model distinguishes between disabilities and impairments, focusing on social barriers that limit the accessibility of certain individuals, ultimately leading to disability (it is not the individual who is disabled; it is the environment that is disabling) [5].

This ongoing debate is part of a broader shift in awareness of the value and rights related to 'difference' within human society. The trend of promoting inclusivity and diversity as core values has a complex and evolving history, with its roots stretching back several centuries. However, it gained significant momentum and formal recognition during the latter half of the 20th century, particularly with the rise of civil and human rights movements. These movements have addressed a wide range of issues, including racial and ethnic discrimination, gender inequality, workers' rights, LGBTQ+ rights, Indigenous rights, and political freedom. The increasing interconnectedness of the world through the internet and social media has also provided a platform for marginalized voices, facilitating the dissemination of information and awareness about social inequalities.

In response to globalization, changing demographics, and societal expectations, many corporations, educational institutions, and governmental bodies began implementing diversity and inclusion initiatives to promote equal opportunities and representation.

Such historical developments, alongside research progression, have led to a transition in the conceptualization of human diversity and a growing recognition and acceptance of neurodiversity. This entry takes a neuroscientific lens on neurodiversity, exploring neurodiversity between/within individuals, evidence of neurodiversity in the brain, evolutionary perspectives on neurodiversity, and neurodiversity in light of the medical and social models of disability.

### 1.2. A Neuroscientific Perspective on Neurodiversity

While neurodiversity rose as a qualitative social concept, a semantic analysis of the term suggests a quantitative, scientific meaning attached to it. The word neurodiversity is composed of the word *Neuro*, which means nervous system and the word *diversity* which means a variety (e.g., various kinds or trait expressions measured in a sample). To deepen our understanding of neurodiversity, it is worth asking what neurodiversity might mean neuroscientifically.

## 2. Neurodiversity between Individuals

Biological measurements typically show a normal distribution [6] where most data points cluster around the mean and fewer data points are located at the two extremes of the distribution. For example, the global average human female height is ~162 cm with ~6 cm per standard deviation. This means that most females are between 153 cm and 171 cm tall, and this range is considered statistically *typical*. Only 20% are shorter or higher than that range and considered statistically *atypical*. Notably, typical and atypical are neutral terms; they are not good or bad but simply imply a probability of expression.

Similarly, human brain characteristics and expressions vary across individuals. Brain characteristics can be an anatomical trait (e.g., structure or size of a brain region), a functional trait (e.g., the activity level of a brain system), or a combination of anatomy and function such as connectivity structure within and between brain systems. Since brain structure and function underly human perception and behavior, it is reasonable to think of the various expressions of human cognition and communication patterns as neurodiversity. While neurodiversity is often utilized to identify and label a deviation from the typical representation of neuronal and behavioral trait(s), it is important to recognize that it represents a statistical description of the population as a whole and effectively, neurodiversity exists across the entire population.

Neuronal phenotypes are better described as continuous than categorical (hence, Autism *Spectrum* Disorder), yet the neurodiversity framework distinguishes between two

main types; *neurotypical* refers to an individual whose brain and cognitive development falls within the typical range. *Neurodivergent* refers to an individual whose brain and cognitive development falls outside (or 'diverges' from) the typical range (see Figure 1).

## Neuronal/Behavioral Trait Expression Across Population

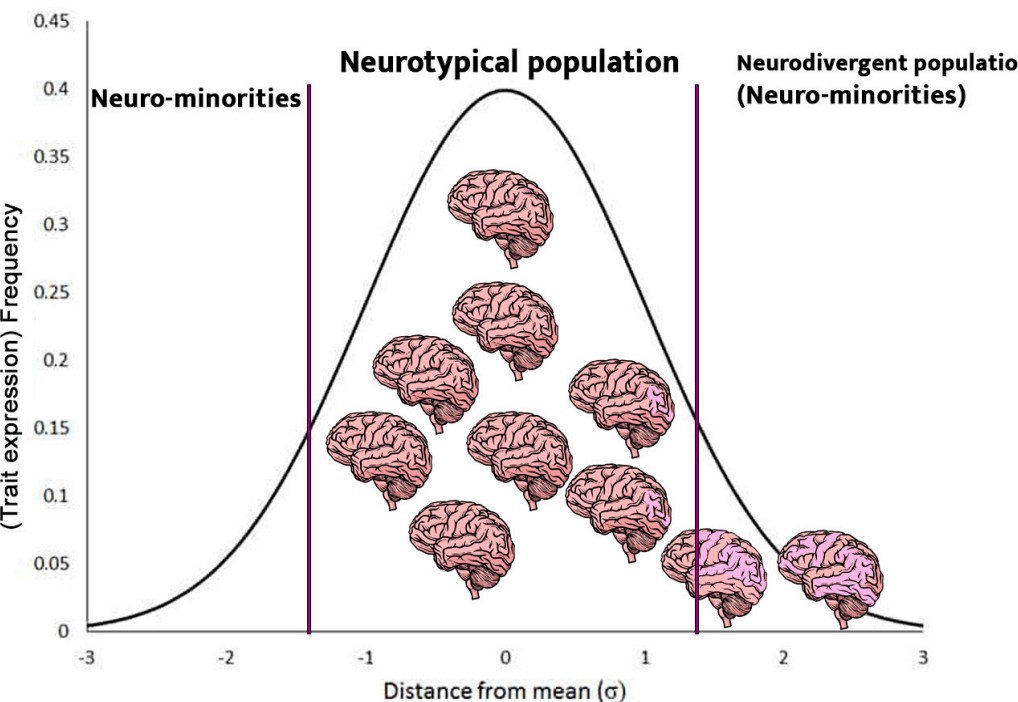

**Figure 1.** Neuronal and behavioral trait expressions are often normally distributed across the population. Neurotypical refers to the majority of the human population that exhibits common, typical neuronal phenotypes that fall close to the mean of the Gaussian. Neurodivergent refers to a minority of the population that express statistically less common (atypical) neuronal and cognitive phenotypes that fall at the tails of the Gaussian distribution. Neurodiversity exists across the entire population but is often attributed to neurodivergent populations that express socially evident atypical neurological phenotypes. In this figure, the variation in colors symbolizes the progressive elevation of neurodiversity expression from the neurotypical to the neurodivergent end. While both groups exhibit the purple color, it crosses a specific threshold to signify neurodivergence.

Taking a statistical, scientific lens on neurodiversity helps to reframe the conversation from normality (normal vs. not normal) to prevalence (typical vs. atypical) and facilitates a factual and nonjudgmental discussion. Recently, Nancy Doyle proposed the use of 'neurominorities' as a neutral and statistically accurate umbrella term for atypical neuronal and cognitive phenotypes [7]. A neurominority is a population that shares common elements of neurodivergence shown by a specific symptom cluster and is, therefore, subject to similar challenges and discrimination in facing the neurotypical society.

Doyle's taxonomy of neurominorities includes four major groups (see Figure 2) that share common features of neurodiversity and, specifically, the process leading to the condition (developmental or acquired) and the functional and health consequence of the condition (applied or clinical).

**Neurominorities Classification**

**Developmental - Applied**
Related to the application
of skills in specific domains
with no health implications

Exaxple: learning disabilities
(e.g., dyslexia, dysgraphia, dyscalculia)

**Developmental - Clinical**
Related to atypical behavior
and communication patterns
across domains.
Considered health issues.

Example: ASD, ADHD

**Acquired - Clinical**

**Neurologic**
Related to functional changes due to illness
or injury to the nervous system.
Considered health issues.

Example: Aphasia, Amnesia

**Psychiatric**
Related to mental disorders affecting
mood, perception and behavior.
Considered health issues.

Example: Depression, schysophrenia

**Figure 2.** The classification of neurominorities (based on Doyle 2020) considers the major process leading to (developmental or acquired) and the functional and health implication (applied or clinical) of the neurodiversity.

I.     Applied developmental neurominority (including learning disabilities, such as dyslexia, dysgraphia, and dyscalculia); neurodevelopmental conditions with genetic and environmental components leading to educational-applied consequences but not health implications.

II.    Clinical developmental neurominority (including Tourette, Autism Spectrum Disorder (ASD), and Attention Deficit and Hyperactive Disorder (ADHD)); neurodevelopmental conditions with genetic and environmental components that impact behavior and communication across contexts and are currently considered health issues.

While both groups refer to neurodevelopmental conditions, the first (applied) typically affects specific domains (e.g., educational settings), and the latter (clinical) has a broad and cross-domain impact.

In addition to developmental neurodiversity, Doyle differentiates between psychiatric and neurological acquired neurominorities, both are clinical and considered health issues.

III.   Psychiatric acquired minority refers to mental illness and psychiatric disorders.

IV.    Neurological acquired minority referees to neurodivergence due to neurological illness or injury.

### 3. Where Is Neurodiversity in the Brain? (Diversity of the "Neurotypical")

The human brain has evolved through a lengthy evolutionary scaffolding leading to specific neuronal constructs and organizations [8,9]. This evolutionary protocol is the blueprint of all human brains, which are eventually more similar than different in their structural and functional organization [10,11].

A closer look reveals that functional brain systems are not equal in the level of inter-subject variability (the difference in brain function between individuals). In fact, specific brain systems showing higher inter-subject variability are responsible for most of the natural variability between individual brains (neurodiversity) [11].

Interestingly, inter-subject variability is correlated to evolutionary cortical expansion. The highest variability was found in frontal, temporal, and parietal association cortex areas, phylogenetically late-developing regions which are essential to complex, human-specific cognitive functions like reasoning, attention control, and language [10,11]. This finding suggests that neurodiversity reflects human individualization; brain systems that facilitate high-order survival functions (such as self-reflection, social processing, and complex decision-making) require environment-specific fine-tuning and, therefore, are more susceptible to experience-based learning.

## 4. Neurodiversity within Individuals

Although the clinical and research community strives for more neurophysiological measurements, the diagnostic process of clinical conditions related to neurodiversity (such as ASD, ADHD, and mental illness) is based on cognitive and behavioral much more than neurological measurements. For example, the diagnosis of ADHD is based on psychoeducational and cognitive scales and not on brain scans. The assessment tools often measure the person's perception and performance in relevant cognitive, emotional, and social domains.

Generally, neurominorities and neurotypicals show different profiles in the psycho-cognitive and executive functions assessments (e.g., IQ test including measurements of spatial, visual, verbal, memory skills, and more). While neurotypicals tend to show a consistent level of (high/low/average) performance across variables, neurodivergent individuals often demonstrate a 'spiky' profile, with high inter-variability in cognitive performance compared to the neurotypical majority [7,12]. A group of individuals with a specific symptom cluster and often a specific *spiky* profile can be considered a *Neurominority* with shared neuronal and perceptual characteristics [7].

For example, people with dyslexia often score high on visual reasoning while scoring low on working memory and processing speed [13]. Moreover, people with dyslexia tend to perform better in global abstract and spatial reasoning than neurotypicals [14]. In addition, the research points to a higher heterogeneity across neurominorities and their subtests compared to the neurotypical population [12,15].

The scientific lens on neurodiversity offers instrumental evidence to advance the social discussion about neurodiversity. In this sense, the high variability within the neurodivergent population suggests that the binary classification of neurotypical and neurodivergent is insufficient to understand and support neurominorities.

## 5. Social Normalizing of Neurodiversity (or the Neurotypical Threshold)

Given that humans are complex, with multiple traits underlined by neurological structures and mechanisms, neurodiversity is multidimensional, and the distinction between 'neurotypical' and 'neurodivergent' is not straightforward. For example, left-handedness is related to atypical brain lateralization that characterizes only 10–15% of the global population and therefore is a form of neurodiversity [16]. While left-handed people suffered discrimination in the past due to their atypicality, this has changed over the years. Today, left-handedness is no longer perceived as sinister and usually does not lead to social exclusion. Hand dominance has been normalized so much that left-handed people are not part of the neurodiversity movement. The case of left-handedness is an example of the link between neurodiversity and dynamic social norms. In many aspects, this normalization process is what the neurodiversity movement is seeking today.

## 6. Evolutional Perspective on Neurodiversity

Most neurodiversity-related conditions are developmental, with genetic and environmental (learning by experience) components affecting the trajectory and individual manifestation of neurodiversity [17]. Developmental and psychiatric neurodiversity (neurominority groups 1–3) have a high heritability rate and are usually not the result of new random mutations [17,18]. The evolutionary preservation of these atypical phenotypes challenges the idea that they merely reflect an error in production and dysfunction.

Research evidence points to a genetic association between the evolution of human-specific cognition and human-specific brain disorders [19]. In other words, neurodiversity is a double-edged sword, with disadvantages (mainly to the individual) and advantages (to the individual and the human species). According to this evolutionary perspective, neurodiversity is a type of biological altruism; individually impairing genetic combinations can carry specific benefits for society. For example, ADHD genetics is associated with novelty-seeking and risk-taking behaviors that elevate individual vulnerability (higher than average risk to physical and social harm and mental illness). At the same time, these behavioral patterns promote resilience at the population level by enhancing diversity,

flexibility, and evolvability. While costs are borne mainly by the individual, and benefits accrue to the entire group [20].

Neurodiversity often offers competitive advantages to the individual on top of the societal benefit [21]. For example, people with high-functioning ASD face multiple difficulties (e.g., in social communication, theory of mind, flexibility, and hyper-sensitivity) but also benefit from some cognitive strengths and advantages, for example, in memory and attention to detail and pattern detection [17].

A new theory highlights human collective cognition as a central force in the resilience and evolvability of the human species [22]. Collective cognition relies on human collaboration and human diversity. Through communication and cooperation, human society benefits from diverse and complimentary cognitive (e.g., learning and problem-solving) strategies that enhance its ability to adjust and thrive in multiple and dynamic environments [22].

## 7. Neurodiversity in Light of the Medical and the Social Models of Disability

The framework used to approach disability has changed significantly over the last half a century, as reflected by the World Health Organization (WHO) Disability Classification Manuals [23–26]. The classic medical model approached disabilities as impairments and dysfunction of the person [23]. The later social model framed disability as the misalignment between the individual (ability and needs) and their environment (accessibility and accommodation capacity) [24]. The progression from the medical to the social model expresses a shift from an illness-based to a health-based approach to disability. Importantly, it reflects a change in basic assumptions from disability as a personal problem to disability as a social construct.

The neurodiversity discussion that started as a grassroots movement, driven by individuals' lived experiences, penetrated the scientific community and evoked the rethinking of human diversity. The autism researcher Baron-Cohen suggested updating medical terms to consider neurodiversity and a more precise differentiation between *disorder* and *disability*, often used interchangeably [17]. According to Baron-Cohen, a disorder is maladaptive functioning in organic and behavioral levels which persist across all environments and contexts. For example, severe anxiety or anorexia are disorders as they have organic and behavioral components and result in severe disturbance of function in any environment.

*Disability* is a disruption of behavioral ability due to internal or external barriers (or their combination) leading to a compromised performance in specific functions and specific environments. For example, dyslexia imposes barriers in specific learning domains and could become a learning disability.

The distinction between the categories is essential, especially in that often neurominorities are labelled with disorders (e.g., Autism Spectrum Disorder and Attention Deficit Hyperactive Disorder). The difference between *disorder* and *disability* could be demonstrated using the example of dyslexia. Similar to disorders, dyslexia has an organic (neuronal) component associated with atypical neuronal processing related to the reading network. However, unlike disorders which apply across contexts, dyslexia often affects a specific (reading-related) domain and does not hamper other learning domains or overall intelligence [27]. Moreover, the extent to which dyslexia imposes a learning barrier greatly depends on the learning environment, resources and tools offered to the students [28,29]. Given the appropriate accommodation, students with dyslexia can be successful learners, which goes against the concept of disorder and even questions the standard classification of a learning disability.

Medical terminology concerning neurodiversity is not merely semantic, as it calls for different actions by policymakers and systemic solutions. Baron-Cohen suggests that while a disorder requires a cure or treatment, disability requires societal adjustments and support. Finally, some cases of neurodiversity reflect an atypical but natural *difference* and mainly require social acceptance and inclusion.

## 8. Toward a New (Scientific and Humanistic) Approach to Neurodiversity

Over the last two decades, neurodiversity has become a topic of interest and has been explored through multiple lenses and disciplines, such as social science, neuroscience, evolutionary science, and medicine. The accumulating body of neurodiversity research and voices within the neurodiversity community challenge the classic medical view of cognitive variations as pathologies (disorders/disabilities) [7,17].

The neurodiversity approach suggests a more dynamic and personalized process in classifying atypical phenotypes into clinical or non-clinical categories (e.g., *disorder*, *disability*, or *difference*). This classification is highly based on the interaction between the individual and their environment. For example, while low-functioning autism could be considered a disorder that calls for treatment, high-functioning autism could be considered a disability or merely a difference depending on the environment's capacity to include and integrate various human needs, styles, and talents.

The neurodiversity approach also highlights the subjectivity of atypicality; while some individuals experience it as a disadvantage, some embrace it as a core identity component. Therefore, the discussion and research on neurodiversity also raise an important question regarding *who* should diagnose or classify neurodiversity into medical and social categories [30]. Traditionally this was the sole authority of medical professionals. However, considering the social and subjective aspects of the neurodivergent experience, a more holistic approach that includes the perspectives of both medical professionals and patients and integrates objective scientific findings with the subjective human experience may lead to more beneficial classifications and health-promoting outcomes for neurominorities.

## 9. Conclusions

Neurodiversity describes the rich variation in human cognitive, sensory, and communication experiences and has emerged as a captivating subject of study across several social and life sciences academic disciplines. Each of these disciplines offers valuable insights into the multifaceted nature of neurodiversity, but it is through an interdisciplinary approach that we can fully comprehend this phenomenon. While considerable progress has been made in understanding neurodiversity, there is still much to be explored and discovered about the genetic and neurodevelopmental basis of neurodivergence, which undoubtedly contributes to understanding the biological roots and functional meanings of neurodiversity.

Educating professionals and the general public on neurodiversity is essential, especially given its high prevalence, with approximately 15 to 20 percent of the population exhibiting some form of neurodivergence. Unfortunately, despite its ubiquity, there remains a lack of widespread knowledge about neurodiversity, even among neurodivergent individuals, leading to misconceptions and stigmatization of neurominorities.

This entry introduces neuroscience insights on neurodiversity and integrates them with humanistic perspectives on neurodiversity. It emphasizes the importance of inclusivity and accommodation, not merely as group-level compromises to aid individuals with special needs but as a means to harness the strengths and potential inherent in neurodiversity for the benefit of the entire population. The humanistic approach highlights the shared humanity across the cognitive range, while the scientific approach links neurodiversity to group-level high performance, adaptability, and resilience. Together, these approaches support a strength-based view of neurodiversity as an enriching and nurturing component of human society.

As societies embark on this journey towards inclusivity, it becomes their inherent responsibility to enable diversity to manifest. It is not enough to passively acknowledge differences; rather, it requires active effort and commitment to dismantle barriers by incorporating universal and flexible designs that embrace and accommodate diverse human needs and ways of expression. By doing so, we invite diverse players to actively participate, contribute, and thrive within the fabric of society.

In conclusion, providing evidence-based knowledge about neurodiversity is pivotal in breaking down stigma, biases, and social barriers imposed on neurominorities. By acknowledging the ongoing research and adopting an interdisciplinary perspective, we can advance our understanding of neurodiversity and work towards creating more inclusive policies in the workplace, medical, and educational systems.

**Funding:** This research received no external funding.

**Institutional Review Board Statement:** Not applicable.

**Informed Consent Statement:** Not applicable.

**Data Availability Statement:** Not applicable.

**Conflicts of Interest:** The authors declare no conflict of interest.

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
