# Peer review of "Unraveling Neurodiversity: Insights from Neuroscientific Perspectives"

_encyclopedia, doi:10.3390/encyclopedia3030070_

Round 1

Reviewer 1 Report

The authors proposed to describe a crucial topic, writing in a clear and simple way, and in the same time, given a scientific and rigorous approach.

The concept of neurodiversity, and the related notions of neurodivergent and neurotypical, is addressed from different points of view, from the historical narrative point of view to the social and neuroscience fields, giving a comprehensive view of the issue.

Finally, the bibliography is appropriate and well cited throughout the text.

I propose to accept this work in the present form for the publication.

Author Response

I thank the reviewer for the thoughtful and positive feedback on the manuscript. This paper aims to offer readers (of various backgrounds) a comprehensive view of neurodiversity in a clear and communicative way, and I am pleased to hear that the efforts have been successful in achieving that. The reviewer’s kind words regarding the clarity and simplicity alongside the scientific rigour are genuinely appreciated. I am grateful for the reviewer’s evaluation and positive assessment of the manuscript.

Reviewer 2 Report

This entry on neurodiversity is a strong contribution to the literature. Thera are minor writing points, and I have some substantive comments. 1. Abstract: “our society” use “contemporary society”. 2. Line 50: “linguistic analysis” use “sematic analysis”. Line 46, top of page 2: “individuals which cause disability”; obscure, improve the writing. Lines 76, 77. Replace “is” by “refers to”. Line 95. ADHD, not  ADHA. Page 3 figure caption. Here and elsewhere, replace “which” by “that “ when there is no preceding comma. “Gaussian” – add “distribution”. Line 135. “therefore are more” add “are”. Line 165. Same comment as for “as” below. Use other words for “since” when “because” is the meaning. Line 171. “travesty” use “sinister or abnormal”. Line 184. “mistake” use error or abnormality”.  Line 232. Replace “as” here and elsewhere when used as “because”. I like using “in that”. Line 244 “merely semantic” add “merely”. Some paras are not indented. Some paras contain one sentence only. Some spelling is Canadian, (I am Canadian, too) – use “behavior”.

Substantively, I find the opposition between scientific and humanistic problematic. (e.g., line 269) Is the author referring to social constructionism? Even so, humanism and social constructionism are both scientific. Perhaps the author means reductionism for the latter.

The argument that neurodiversity applies to everyone is a bit of a stretch -granted, there are individual differences in everyone, but does that mean neurodiversity? The argument seems to undermine the purpose of the term. Perhaps add a qualifier here

The entry allows one to ponder whether more inclusive terminology should be used than neurodiversity, such as neuro(bio)diversity or neuro-environmental diversity, or a combined term, but this could be too clumsy.

I did a quick PsychInfo search, and here are 2 articles that might help in this regard

Annual research review: Shifting from ‘normal science’ to neurodiversity in autism science
Pellicano, Elizabeth; den Houting, Jacquiline.  Journal of Child Psychology and Psychiatry Vol. 63, Iss. 4,  (Apr 2022): 381-396.

Applied behavior analysis and the abolitionist neurodiversity critique: An ethical analysis
Graber, Abraham; Graber, Jessica.  Behavior Analysis in Practice (Mar 2, 2023).

This entry on neurodiversity is a strong contribution to the literature. Thera are minor writing points, and I have some substantive comments. 1. Abstract: “our society” use “contemporary society”. 2. Line 50: “linguistic analysis” use “sematic analysis”. Line 46, top of page 2: “individuals which cause disability”; obscure, improve the writing. Lines 76, 77. Replace “is” by “refers to”. Line 95. ADHD, not  ADHA. Page 3 figure caption. Here and elsewhere, replace “which” by “that “ when there is no preceding comma. “Gaussian” – add “distribution”. Line 135. “therefore are more” add “are”. Line 165. Same comment as for “as” below. Use other words for “since” when “because” is the meaning. Line 171. “travesty” use “sinister or abnormal”. Line 184. “mistake” use error or abnormality”.  Line 232. Replace “as” here and elsewhere when used as “because”. I like using “in that”. Line 244 “merely semantic” add “merely”. Some paras are not indented. Some paras contain one sentence only. Some spelling is Canadian, (I am Canadian, too) – use “behavior”.

Author Response

I would like to express my sincere gratitude to the reviewer for the thorough and thoughtful review of the paper. The reviewer’s generosity in dedicating time and effort to provide such detailed and meticulous feedback is truly appreciated. The reviewer's suggestions were implemented in the text and enriched the content and presentation of the paper. Please see the attached document for a detailed description of how the comments were addressed in the revision process.

Reviewer 3 Report

Thank you for considering me to review your entry in the encyclopedia. The contribution is titled "Neuroscientific View on Neurodiversity," which accurately reflects the author's objective to explore the neurogenetic perspective of neurodiversity. In recent years, research in the fields of neuroscience, psychology, and genetics has yielded increasing evidence supporting the existence of a diverse range of neurological differences and their associated strengths. This entry has the potential to play a crucial role in further solidifying the emerging "paradigm change" surrounding neurodiversity. I have a few minor suggestions for the authors to consider, primarily focused on enhancing the flow of ideas and providing readers with a clear "navigation tool." Nevertheless, it is an excellent discourse that is a pleasure to read.

TITLE

I think the title represents a snapshot of what this entry in the encyclopedia is all about. Well done. Please note that since the mission here is to focus on the neuroscientific view or a neurogenetic perspective, I would encourage the authors to first introduce the factors that fostered the emerging view of neurodivergent and later only focus on neurogenetic evidence of neurodivergent. The social movement or other precursors should be introduced in the first paragraph. These suggestions will be recapitulated below in detail.

Minor issue: Maybe Neuroscientific Views on Neurodiversity

SUMMARY/SYNOPSIS

Minor suggestion for something akin to synopsis/definition. I think these statements (“What does neurodiversity mean scientifically” should be continued with the following narration.  Suggestion: “The present discourse review focusses on existing scientific evidence on neurodiversity between and within individuals, evidence of neurodiversity in the brain, and an evolutional perspective on neurodiversity.  In order to lay the groundwork for the present entry, the literature will be viewed on the social normalisation of neurodiversity and neurodiversity in light of the medical and social models of disability”. 

WRITEUP

The author did provide focus narration, which was demarcated into different subheadings.  However, there is little flow of ideas between the paragraphs.   Therefore, I have the following suggestion.  In a paragraph similar to the introductory,  the authors should cover several factors that have contributed to its growing recognition and acceptance of neurodiversity, including (1) the advocacy and Activism movement that has challenged societal stigma, promote inclusivity, and fight for the rights and acceptance of neurodiversity individuals, (2) Paradigms of Disability where the medical model of disability has been deemed to ‘pathologise’ neurological differences; (3) Autistic advocacy where such neurodivergent individuals have been at the forefront of the eroded biomedical model;  (4)  Other factors that came to the forefront in efforts to create inclusive educational environments and employment opportunities within the neurodiversity perspective.  These points should be mentioned as an introduction, and this implies this subheading (1. Different Perspectives on Neurodiversity) should change to 'historical development of the concept of neurodiversity or along these terms’.  

At the end of an introductory paragraph, the authors should define the topics to be covered, namely, different perspectives on neurodiversity, neurodiversity between/within individuals, evidence of neurodiversity in the brain, evolutional perspectives on neurodiversity, and neurodiversity in light of the medical and the social models of disability. 

CONCLUSION

I believe that the conclusion should be reconsidered. It would be beneficial to acknowledge that the neurogenetic aspect of neurodiversity is still in its early stages. Furthermore, the authors should provide a conclusive assessment regarding the heuristic value of the present perspective on neurodiversity, considering studies on variations between and within individuals, evidence of neurodiversity in the brain, and an evolutionary perspective on neurodiversity.

Lastly, it would be valuable for the authors to address a potential counterview to neurodiversity, which suggests that while neurological evidence supports the concept, there is an argument that behavioural changes can influence brain connectivity and, in turn, the brain can shape behaviour. This raises the concern that the entire paradigm could be seen as tautological.

Overall, this entry is intriguing and makes a valuable contribution to the encyclopedia.

Author Response

I want to express my sincerest gratitude for the reviewer's profound and eye-opening insights that have been instrumental in elevating this manuscript to the next level. The feedback has been invaluable in crystallizing the essence of this entry paper and making it more coherent, cohesive and communicative. Please see the attached file for a detailed description of how the comments and suggestions were implemented in the revision process.

Round 2

Reviewer 2 Report

2nd mention of disorders should use abbreviations

Author Response

I thank the reviewer for the feedback and appreciate the attention to detail and the opportunity to address the comment.

Regarding the use of abbreviations, the terms were introduced along with their abbreviations upon their first mention in the manuscript. Subsequently, abbreviations are consistently used throughout the text, except in one case where the full term is essential for explicit contextual understanding. In the paragraph discussing the controversy surrounding medical labelling of certain conditions as disorders (lines 254-255), I found it necessary to include the full term to ensure the precise communication of the point at hand (that while there is a debate about whether some conditions are disorders, they are labelled as such). I believe this is the only place where the full terms were used instead of abbreviations.

Once again, I genuinely appreciate your valuable feedback to ensure that the manuscript adheres to the best practices in abbreviation usage while ensuring clarity in conveying the intended message.

Thank you for your time and consideration.

Reviewer 3 Report

Thank you for resending the revised manuscript titled "Unraveling Neurodiversity: Insights from Neuroscientific Perspectives." I'm pleased to see that the authors have thoughtfully incorporated my comments, resulting in an extensive revision of the manuscript. As a result of her hard work, the manuscript has now reached a level of improvement that justifies its publication in the Encyclopedia.

Author Response

Thank you for your comments.

Round 3

Reviewer 2 Report

accept as is